# Gene Expression Profiles in Two Razor Clam Populations: Discerning Drivers of Population Status

**DOI:** 10.3390/life11121288

**Published:** 2021-11-24

**Authors:** Heather A. Coletti, Lizabeth Bowen, Brenda E. Ballachey, Tammy L. Wilson, Shannon Waters, Michael Booz, Katrina L. Counihan, Tuula E. Hollmen, Benjamin Pister

**Affiliations:** 1Southwest Alaska Network, Inventory & Monitoring Program, National Park Service, Fairbanks, AK 99709, USA; 2Western Ecological Research Center, U.S. Geological Survey, Davis, CA 95616, USA; lbowen@usgs.gov (L.B.); swaters@usgs.gov (S.W.); 3Alaska Science Center, U.S. Geological Survey, Anchorage, AK 99501, USA; beballachey@gmail.com; 4Massachusetts Cooperative Fish and Wildlife Research Unit, University of Massachusetts, U.S. Geological Survey, Amherst, MA 01002, USA; tammywilson@umass.edu; 5Alaska Department of Fish and Game, Homer, AK 99603, USA; michael.booz@alaska.gov; 6Alaska SeaLife Center, Seward, AK 99664, USA; katrinac@alaskasealife.org (K.L.C.); tuulah@alaskasealife.org (T.E.H.); 7Eastern Regional Research Center, U.S. Department of Agriculture, Wyndmoor, PA 19038, USA; 8College of Fisheries and Ocean Sciences, University of Alaska Fairbanks, Seward, AK 99664, USA; 9Kenai Fjords National Park, National Park Service, Seward, AK 99664, USA; Benjamin_Pister@nps.gov

**Keywords:** Pacific razor clam, *Siliqua patula*, gene expression, environmental drivers, clam population decline, predation

## Abstract

With rapidly changing marine ecosystems, shifts in abundance and distribution are being documented for a variety of intertidal species. We examined two adjacent populations of Pacific razor clams (*Siliqua patula*) in lower Cook Inlet, Alaska. One population (east) supported a sport and personal use fishery, but this has been closed since 2015 due to declines in abundance, and the second population (west) continues to support commercial and sport fisheries. We used gene expression to investigate potential causes of the east side decline, comparing razor clam physiological responses between east and west Cook Inlet. The target gene profile used was developed for razor clam populations in Alaska based on physiological responses to environmental stressors. In this study, we identified no differences of gene expression between east and west populations, leading to two potential conclusions: (1) differences in factors capable of influencing physiology exist between the east and west and are sufficient to influence razor clam populations but are not detected by the genes in our panel, or (2) physiological processes do not account for the differences in abundance, and other factors such as predation or changes in habitat may be impacting the east Cook Inlet population.

## 1. Introduction

Alaska’s Pacific razor clams (*Siliqua patula*) are important for commercial and personal harvest and as prey for marine animals [1,2,3]. Currently, the only commercial razor clam fishery in Alaska occurs in west Cook Inlet (WCI) near Polly Creek, and the annual harvest has averaged approximately 900,000 clams since 1980 [3,4]. Recreational clam harvests occur more widely in Cook Inlet and provide a boost to local economies [5]. The state’s largest sport and personal use Pacific razor clam fishery historically occurred along a 50-mile area of beach between the Kasilof and Anchor rivers on the east side of Cook Inlet (ECI), where an average of almost one million clams per year were harvested from 1977–2006 [6], similar to the annual commercial harvest that is ongoing on WCI. Razor clam harvest was not evenly distributed throughout the ECI area and primarily occurred on the Clam Gulch and Ninilchik area beaches. This fishery remained stable during this period with consistent recruitment of new age classes (juveniles) to the beaches, and harvest was comprised of a broad range of age classes on all beaches [6]. However, between 2009 and 2012, the annual number of clams harvested per digger declined by 41% concurrent with a decline in harvest effort below the long-term mean (1977–2008), indicating a dramatic decline in the ECI razor clam population [3,7]. During this period of decline, average annual age and length compositions of the harvest were truncated to younger/smaller clams compared to historical averages and were comprised of fewer cohorts on all beaches regardless of harvest rates [8]. The human exploitation rate of razor clams throughout most of ECI was assumed to be low based on digger distribution and monitoring at more heavily harvested beaches at Clam Gulch and Ninilchik [9]. The clam decline resulted in restrictions of ECI clamming in 2013 with full closure in 2015; however, ECI razor clams have not recovered to historical abundances [10,11]. The causes of the decline through 2015 were unknown but were attributed to poor recruitment into the adult age classes and above average adult natural mortality [12]. In contrast, the razor clam fisheries in WCI have continued to support a commercial and personal use fishery over the same time frame [3].

East Cook Inlet beaches are accessible by road, and personal harvest pressures on razor clams there have been greater than in WCI, which is accessible only by boat or plane. Based on monitoring data, clam abundance among locations in ECI is not uniform [3], suggesting that razor clam populations are structured at small spatial scales. For example, recent surveys determined that at Ninilchik, juvenile clam mortality was 70%, while at Clam Gulch it was only 10% [3]. Razor clam recruitment is also variable among locations and across years with, for example, more consistent recruitment at Clam Gulch and less frequent, larger recruitments at the Ninilchik beaches [3]. Razor clam growth rates also differ among ECI locations, with increasing growth rates from north to south [6,13]. WCI razor clam data are primarily limited to commercial fishery data, which include digger effort (digger days), number harvested, and pounds harvested annually. The commercial harvest makes up over 97% of the total razor clam harvest annually in WCI, indicating the personal use and sport fishery is minimal [3].

Although causes for the decline of razor clams in ECI are unknown, there are potential differences beyond any effects of harvest in drivers between ECI and WCI. Geomorphological conditions vary between ECI and WCI and may contribute to differences in the quantity and stability of habitats [14]. However, ShoreZone^®^, a mapping and classification system that specializes in the collection and interpretation of imagery of the coastal environment, indicates almost no difference in sediment type, wave exposure, or oil residence index across all sites [15]. Oceanographic processes in Cook Inlet contribute to seasonal and spatial variation in salinity and temperature [16] as well as plankton blooms [17]. Higher temperatures, salinity levels, and the presence of contaminants have been shown to suppress bivalve immune function, making them more susceptible to disease and parasites, potentially resulting in more severe infections [18,19]. More recent studies have highlighted food web complexity differences between ECI and WCI, likely due to the influence of glacial melt in the west, an influence which is absent in the east [20]. Additionally, top-level predators, including brown bears (*Ursus arctos*) [21] and sea otters (*Enhydra lutris*) [22] may be a factor, and if so, predation pressures will vary between ECI and WCI likely due to predator population density differences across the study region [23,24].

Increasingly, gene expression is being used as a tool to monitor nearshore marine ecosystems [25,26,27]. As an emerging field, gene expression focuses on mussels as sentinels of ocean health [28,29], but methods are now widely applied across marine species, including invertebrates, mammals, birds, and fish [30,31,32,33]. These gene-based health diagnostics provide an opportunity for an alternate, holistic assessment of health, not only in individuals or populations, but potentially that of ecosystems [34]. Specifically, gene expression is the process by which information from the DNA template of a particular gene is transcribed into messenger RNA (mRNA) and eventually translated into a functional protein. The amount of a particular gene that is expressed is physiologically dictated by the number of intrinsic and extrinsic factors, including stimuli such as nutrient levels, toxin exposure, or changes in water chemistry or temperature. Expression analysis can effectively provide an early warning system for pathophysiological changes within an organism exposed to biological or physical stressors, as altered levels of gene expression will be evident prior to clinical manifestation [26,35,36]. The comparison of gene expression patterns identifies genes that are differentially expressed in distinct populations or in response to different treatments or exposures. For these reasons, we employed gene expression to improve our understanding of drivers that may influence razor clam populations in Cook Inlet.

In a previous study [37], we used gill tissue to identify a target gene expression profile for Pacific razor clams, based on known physiological responses to environmental stressors in other invertebrates (Table 1). For example, increased ocean acidification leads to increased levels of calmodulin (CaM) expression and would result in changes in metabolism and immune response [38,39]. Pathogen exposure likely leads to changes in levels of ferritin (Ferr) [40,41] and Peptidylprolyl isomerase A (PPIA) expression and would result in an inflammatory response [42]. Increased expression levels of both heat shock protein 70 (HSP70) and 90 (HSP90) provide cellular protection and are indicative of thermal and general stress (including pathogen exposure), while HSP90 expression levels are also indicators of contaminant exposure [43,44,45,46]. Sampling sites in Bowen et al. [37] were located within two national parks and considered to be pristine. However, our panel was able to identify significant differences in gene expression between parks and among sites within each park, which indicated variation in both large-scale and local environmental conditions.

Our objectives in this study were to: (1) determine gene expression profiles for Pacific razor clams from two areas, ECI and WCI, (2) assess differences in expression between the two areas, as well as among sites within each area, and (3) based on any differences (or lack thereof) that may be observed, provide insight as to possible drivers of the ECI razor clam decline. We hypothesized that the comparison of gene expression profiles between ECI and WCI would help differentiate between drivers affecting Pacific razor clams in ECI. If expression profiles differed among areas (or sites within an area), individual genes may suggest processes that have adversely affected razor clams in ECI. Although gene expression studies generally focus on genes that are differentially transcribed among groups, genes that show no difference among groups are also informative. If no expression differences are found, our conclusions can be interpreted in two ways: (1) differing environmental conditions exist but did not influence the molecular physiology reflected by the genes in our panel, or (2) other pressures such as predation or changes in a habitat may be impacting razor clam populations differentially but do not induce a physiological response. Our goal is to provide information to managers to better understand mechanisms limiting recovery of declining Pacific razor clam fisheries in Cook Inlet, Alaska.

## 2. Materials and Methods

### 2.1. Study Organisms

Razor clams were collected in July 2015 and 2016 at nine sites on the east coast of Cook Inlet (ECI; sites north to south: Cohoe, Clam Gulch North, Clam Gulch South, North Oil Pad Access, South Oil Pad Access, Ninilchik North, Ninilchik South, Ninilchik Bar, and Deep Creek) and at three sites along the west coast of Cook Inlet adjacent to Lake Clark National Park and Preserve (WCI; sites north to south: Polly Creek, Silver Salmon Creek, and Chinitna Bay [37]) (Figure 1). Ten clams were collected from each site in 2015 and again in 2016 (total of 120 each year). The ECI sites are monitored annually by the Alaska Department of Fish and Game (ADF&G) while the sites in WCI were selected based on where clam presence had been confirmed by recreational human harvest [37]. Clams were collected at low tide, and sampling was constrained to the same low tide series for both east and west.

### 2.2. Tissue Collection and RNA Extraction

Gill tissue was collected from each clam and placed immediately into RNAlater (Ambion/Life Technologies, Grand Island, NY, USA). All tissue samples were stored at −80 °C. Total RNA was extracted from homogenized gill tissue using the RNeasy Lipid Tissue Mini Kit (Qiagen; Germantown, MD, USA). To remove contaminating genomic (g)DNA, the spin columns were treated with 10 U μL^−1^ of RNase-free DNase I (DNase, Amersham Pharmacia Biotech Inc.; PA, USA) at 20 °C for 15 min. RNA was then stored at −80 °C pending further analyses.

### 2.3. cDNA Synthesis

A standard cDNA synthesis was performed on 2 μg of RNA template from each clam. Reaction conditions included 4 units reverse transcriptase (Omniscript, Qiagen, Valencia, CA, USA), 1 μM random hexamers, 0.5 mM each dNTP, and 10 units RNase inhibitor, in RT buffer (Qiagen, Valencia, CA, USA). Reactions were incubated for 60 min at 37 °C, followed by an enzyme inactivation step of 5 min at 93 °C, and then stored at −20 °C until further analysis.

### 2.4. Primer Design

Primers used were developed by Bowen et al. [37]. Briefly, degenerate primer pairs developed for the razor clam were used on cDNA from three randomly selected clam samples. Degenerate primer pairs were designed to amplify five genes of interest and two reference genes. The PCR amplifications using these primers were performed on 20 ng of each cDNA sample in 50 μL volumes containing 20–60 pmol of each primer, 40 mM Tris-KOH (pH 8.3), 15 mM KOAc, 3.5 mM Mg (OAc)_2_, 3.75 μg/mL bovine serum albumin (BSA), 0.005% Tween-20, 0.005% Nonidet-P40, 200 μM each dNTP, and 5U of Advantage 2^®^ Taq polymerase (Clontech, Palo Alto, CA, USA). The PCR was performed on an MJ Research PTC-200 thermal cycler (MJ Research, Watertown, MA, USA) and consisted of 1 cycle at 94 °C for 3 min, and then 40 cycles at 94 °C for 30 s, at 60 °C for 30 s, and 72 °C for 2 min, with a final extension step of 72 °C for 10 min. The products of these reactions were electrophoresed on 1.5% agarose gels and resulting bands visualized by ethidium bromide staining. Definitive bands representing PCR products of a predicted base pair size of the targeted gene were excised from the gel and extracted and purified using a commercially available nucleic acid-binding resin (Qiaex II Gel extraction kit, Qiagen, Valencia, CA, USA).

### 2.5. Real-Time PCR

Real-time PCR reactions for the individual, razor clam-specific reference genes (18S and Elongation Factor Alpha-1 (EF1a)) and genes of interest were run in separate wells. Briefly, 1 μL of cDNA was added to a mix containing 12.5 μL of QuantiTect Fast SYBR Green^®^ Master Mix [5 mM Mg 2+] (Qiagen, Valencia, CA, USA), 0.5 μL each of forward and reverse sequence specific primers (Invitrogen, Carlsbad, CA, USA), and 10.5 μL of RNase-free water; total reaction mixture was 25 μL. The reaction mixture cDNA samples for each gene of interest and reference genes were loaded into Fast 96-well plates in duplicate and sealed with optical sealing tape (Applied Biosystems, Foster City, CA, USA). Reaction mixtures that contained water, but no cDNA, were used as negative controls. Data are reported as cycle threshold (C_T_) crossing values; lower values indicate higher levels of expression.

### 2.6. Calculation

The stability of both reference genes was evaluated and ranked using the web-based analysis tool RefFinder (https://www.heartcure.com.au/reffinder/; accessed on October–December 2020) [57]. The more stable reference gene was selected for use in normalization of the five genes of interest. We used generalized linear mixed models (GLMM) within the LMER4 package in R version 3.6.1 to test for differences between areas and among sites on the east and west sides of Cook Inlet. The random effect included a site by year interaction to account for site level and interannual variation. Models for each gene expression were fit separately, and we performed post hoc Tukey tests to assess the differences between the means. We also used GLMM to obtain gene expression values for each of the twelve sites, retaining only year as a random effect. We again performed post hoc Tukey tests to make inferences about site-level differences. We show these results using boxplots constructed in R. As neither age nor length affected expression levels in a previous study on razor clams in LACL utilizing the same genes, neither were included in these analyses [37]. We assessed relationships among gene expression levels using Pearson correlations (NCSS, Statistical and Power Analysis Software, Kaysville, UT, USA).

## 3. Results

Over 2 years, approximately 20 Pacific razor clams were collected from each site. The expression levels of five genes of interest and two reference genes were assessed in a total of 230 razor clams. The means and confidence intervals for each site are presented in Table 2. Analyses directed at identifying differences between ECI and WCI, that corrected for inter-site and interannual variation, revealed no differences between the east and west sides of Cook Inlet (Figure 2). Analysis of individual sites revealed significant differences among sites for all genes (Table 2 and Figure 3); however, the magnitude of differences among sites was small, and no consistent patterns were observed. Correlations among the gene expression levels were statistically significant (*p* < 0.05) and positive but low (ranging from 0.16 to 0.71). Relatively strong correlations were found between CaM and PPIA (0.71), and between HSP70 and HSP90 (0.61). No correlation was observed between CaM and either HSP70 or HSP90.

## 4. Discussion

Currently, causes of the Pacific razor clam decline in ECI are not well understood, but the fishery has been restricted or closed since 2013 due to low abundance [13]. Sampling conducted in 2020 indicates that while juvenile recruitment at ECI sites was similar to historical averages, adult mortality across sites was relatively high [11]. In contrast, razor clams in WCI continue to support commercial and recreational fisheries, and although declines in sport, personal and commercial harvests have been documented recently, they appear to reflect lower harvest efforts as opposed to declining stocks [3]. In an effort to understand potential pressures influencing razor clam abundance in WCI and ECI, we used gene expression to assess the molecular responses of razor clams to their environments. Gene expression, which includes the study of transcriptomes and their function (http://www.nature.com/subjects/transcriptomics; [32,58]), is an evolving approach to biomarker monitoring. Little is known about the genomic responses of Pacific razor clams to environmental stimuli. Recently, we completed the first study using gene expression diagnostics as a monitoring tool for the Pacific razor clam, including clams sampled from WCI [37]. Because of recognized and potential differences in clam demographics and population status between ECI and WCI [3], we anticipated finding population differences in gene expression in Pacific razor clams.

Differences among populations are often attributed to a variety of drivers, including those that elicit a physiological response (e.g., parasites, disease, contaminants, nutrient availability), and those that may not (e.g., habitat degradation, predation) [59,60]. For example, parasite infections can reduce growth, overall condition and reproduction, and cause mortality in bivalves [19,61], and parasites including trematodes and *Haplosporidium* spp. have been found in razor clams in Alaska [62]. A study in 2010 identified parasites in many razor clams sampled from a site in ECI (Clam Gulch) [63]. However, parasite levels in Pacific razor clams in WCI have not been reported. Pathogens are also known to play roles in shaping ecosystems [37]. However, nuclear inclusion X (NIX), a pathogen of serious concern in other Pacific razor clam populations in the Pacific Northwest (Washington and Oregon) is rare in Alaska [64] and in fact, NIX was not identified in samples from our sites in a parallel study conducted in 2019 [65]. Contaminants are widely recognized as potentially important stressors on marine ecosystems [66]. However, while not directly measured in razor clam tissues, contaminants are not thought to be an issue in Cook Inlet, based on other invertebrate data from the northern Gulf of Alaska [67].

There are oceanographic (both physical and biological) and geomorphological differences between ECI and WCI [13,16]. The amount and topography of razor clam habitat differs between ECI and WCI, with WCI having significantly larger razor clam beaches. However, based on ShoreZone^®^ sediment type, the exposure and oil residency index values indicate almost no difference between ECI and WCI or among individual sites [15]. The similarity in physical structure and exposure across the sites is to be expected. Unlike mussels, razor clams require specific habitat features to flourish and are not ubiquitous across the north Pacific [1]. Dynamic attributes, such as temperature and salinity gradients, do exist between the east and west sides of the Inlet [16]. The timing of algal blooms, as a source of nutrients, differs between the east and west coasts of Cook Inlet. The major factors responsible for initiating blooms in lower Cook Inlet are water stratification, incident radiation, and water clarity [17,68]. Favorable conditions for the bloom occur first in Kachemak Bay, with a longer residence time and lower mixing rates, permitting surface water to warm in the spring and retain phytoplankton populations at high concentrations [17,68]. The major component of lower east Cook Inlet water originates in the Gulf of Alaska and does not contain the heavy load of suspended inorganic particles present in the upper Cook Inlet water which dominates the western side.

With differential environmental pressures such as these, we might have expected to see variation in expression of the target genes between WCI and ECI. However, we found no significant differences in expression levels of genes between WCI and ECI when we compared the areas with sites grouped together. Furthermore, when we compared expression levels from WCI and ECI with levels from Pacific razor clams in Katmai National Park (samples collected in same season and year; data from [37]), clams from Katmai had significantly higher expression of CaM, PPIA, and Ferr, and significantly lower expression of HSP70 than clams from WCI or ECI. The implications are that WCI and ECI are relatively similar in terms of environmental drivers such as pathogens, ocean acidification, contaminants, and nutrient availability pressures, while pressures at Katmai, including pathogens and nutrient availability, apparently differ. The contrast between razor clams sampled from Katmai with those from other areas (WCI and ECI) supports the diagnostic value of the genes included in our panel. The significant correlations seen among the genes in our panel suggest concurrent responses to environmental conditions. It should be noted that our sampling occurred only in June and July, which are generally months of elevated primary productivity compared to other times of the year [69].

A potentially strong driver of environmental differences within Cook Inlet is the convergence of freshwater discharges and the Alaska coastal current, which supports higher nutrient availability important for growth of sessile organisms [13]. However, ocean current effects and other processes are not necessarily uniform across sites. For example, Ninilchik and Clam Gulch have been identified in previous studies as differing in mortality, recruitment, and growth rates of Pacific razor clams [3,6,13,63]. As discussed by Blackmon [10], bivalve growth rates vary both temporally and spatially, and these patterns may relate to physical and environmental drivers, such as temperature, salinity, sediments, and phytoplankton [70,71,72,73,74,75].

Seasonal distribution of sea ice within Cook Inlet may also play a role in clam population status. Warming ocean temperatures and a rapidly changing climate, particularly in high latitudes, may decrease the seasonal ice present in Cook Inlet. This could result in greater disturbance to shorelines if ice is more mobile and scours the shorelines to a greater extent. Alternatively, with less ice, the shorelines may be more vulnerable to storm surges, impacting habitat, and clam survival. In either case, changes in ice cover could affect single sites, as opposed to Inlet-wide disturbances.

During our sampling period, a marine heatwave (“The Blob”), unprecedented in spatial extent and duration, occurred in the north Pacific [76]. Temperature differences that might have existed at the site level and could contribute to variation in growth or other metrics were likely dampened by this large-scale climactic event [77,78].

When all sites were analyzed individually, we found statistically significant expression differences among genes at all sites (Figure 3). However, the expression differences were small. Varying levels of expression within or among groups or individuals may be normal responses to stimuli and while statistically significant, are likely not biologically relevant [37]. Continuing studies, including controlled exposures, will clarify the biological relevance of differences in gene expression in Pacific razor clams [37,79].

Little is known about the genetic makeup of Pacific razor clams in comparison with other commercially harvested invertebrate species, and our initial research suggested that razor clams are highly genetically divergent from other bivalves. However, the genes targeted in our study have been shown to respond to stressors, both with the closely related Chinese razor clam (*Sinonovacula constricta*) and other bivalves [38,39,40,41,42,43,44,45,46]. In a previous study we determined this target gene panel to be effective at identifying differences between Pacific razor clam populations in seemingly pristine areas [37]. Our targeted expression panel enabled identification of differences in nutrient/energy availability as well as contaminant and pathogen presence between two national parks and among sites [37]. Specifically, the targeted panel enhanced our ability to understand the potential input of both large- and fine-scale processes and pointed to a higher quantity or quality of nutrients at Lake Clark National Park (LACL) in comparison with Katmai National Park (KATM), a higher pathogen response at KATM in comparison with LACL, and a higher response to contaminants at some individual sites than at others.

Although expression studies generally focus on genes that are differentially transcribed among groups, a finding that expression does not vary among groups is also informative. For example, similar gene expression profiles contributed to a body of evidence that pointed to predation as the leading cause of population decline in Southwest Alaska sea otters [80,81]. With no notable expression differences found between WCI and ECI clams, there are alternative conclusions to consider: (1) differing environmental conditions exist that affect expression but were not detected by our gene panel or (2) pressures such as changes in habitat or predation may be impacting razor clam populations in ECI, but do not elicit a physiological response in the clams and therefore would not be detected in this study.

The importance of physical and oceanographic drivers to the small-scale spatial structuring of some intertidal species likely varies with the extent of top-down pressures [82,83]. Possible top-down drivers of Pacific razor clam abundance include human harvest and predation by animals such as brown bears [21,84] and sea otters [22]. Human overharvest has not been implicated in the decline of razor clams in ECI, primarily due to relatively low harvest rates [9] and lack of recovery after several years of fishery restrictions and subsequent closure [3]. Previous studies along the Katmai National Park and Preserve coast, just south of LACL, have observed bears consuming intertidal resources, but the level of predation is not thought to be sufficient to impact bivalves at the population level [84]. Bear predation is not thought to be a factor driving clam abundance at ECI, as bears are rarely seen on ECI beaches. In contrast, sea otters are widely recognized for their role in structuring nearshore marine ecosystems [85,86,87,88,89,90]. The sea otter is a keystone predator [85] whose historical range encompassed the entire coastal north Pacific before they were almost extirpated, except for small remnant populations, due to the maritime fur harvest [91]. International protections beginning in 1911 have allowed sea otters to reoccupy habitats [92] where invertebrate populations flourished in their absence. Through predation, a decrease in the abundance of select marine invertebrates is often an outcome of sea otter recolonization, including species important to both commercial, subsistence, and recreational fisheries [93,94,95,96,97,98]. The sea otters’ diet is dominated by a variety of clam species throughout the Gulf of Alaska [59,88,99,100]. Studies conducted in Washington, where sea otters have recently expanded into areas with high densities of Pacific razor clams, report sea otter diets consisting largely of razor clams [22]. The decline of razor clams in ECI may result from the expansion of the sea otter population moving north from Kachemak Bay along ECI (Figure 4). Aerial surveys conducted in 2002 estimated the abundance of sea otters in ECI (excluding Kachemak Bay) to be approximately 962 animals [101]. By 2017, the ECI abundance estimate for sea otters (excluding Kachemak Bay) was 3164 animals [23], a 230% increase. Estimates for the WCI area have also increased during the same time frame (6918 sea otters in 2002; 10,737 in 2017; a 55% increase; [101]); however, in 2017, sea otters had not expanded their range north of Kamishak Bay ([23]; Figure 4), where expanses of razor clam beaches and harvests persist. While not directly measured in this study, the distribution and abundance of sea otters are different between the two sides of Cook Inlet [23] and are potentially drivers in the population status of razor clams that deserve further investigation.

## 5. Conclusions

This study introduces an application of new technology (Pacific razor clam gene expression; [37]) to investigate an ecosystem-level problem. The genes we evaluated generally respond to a wide variety of environmental stressors commonly associated with coastal marine habitats. Although there may be environmental conditions that differ among our study sites, the conditions likely do not account for the divergent population trends in razor clams. Our results suggest that pathogens, contaminants, nutrients, or physiological stress are not driving population abundance at the scale of Cook Inlet. We did not measure direct predation pressure exerted by sea otters on our sampled beaches. Nevertheless, we know that when sea otters newly recolonize an area, significant declines in their invertebrate prey populations will predictably follow. Based on the most recent aerial survey data from Cook Inlet, sea otters have developed a significant presence along the ECI coastline and are virtually absent at our study sites along the WCI coastline [23]. However, surveys flown along the WCI coastline during the summer of 2019 observed higher sea otter abundance, indicating sea otters are moving northward along the west side of Cook Inlet [102]. The contrast in sea otter presence between ECI and WCI suggests they may be a contributing cause of the razor clam decline in the east and if so, suggests a similar outcome for razor clams to the west if sea otter expansion continues.

The extent that various drivers structure razor clam communities at both small and large spatial scales warrants further examination. To improve our ability to use gene expression to attribute specific environmental drivers to detrimental population outcomes for Pacific razor clam and other bivalve species, controlled laboratory exposure studies with whole transcriptome responses are advised. Additionally, consideration of the cascading effects of climate change should complement gene expression methods in evaluating trends in populations. Monitoring the expansion of sea otters as they reoccupy habitat along WCI will be important to determine the role of sea otters in structuring invertebrate populations and aid management of the clam harvest by anticipating and responding to ecological change.

## Figures and Tables

**Figure 1 life-11-01288-f001:**
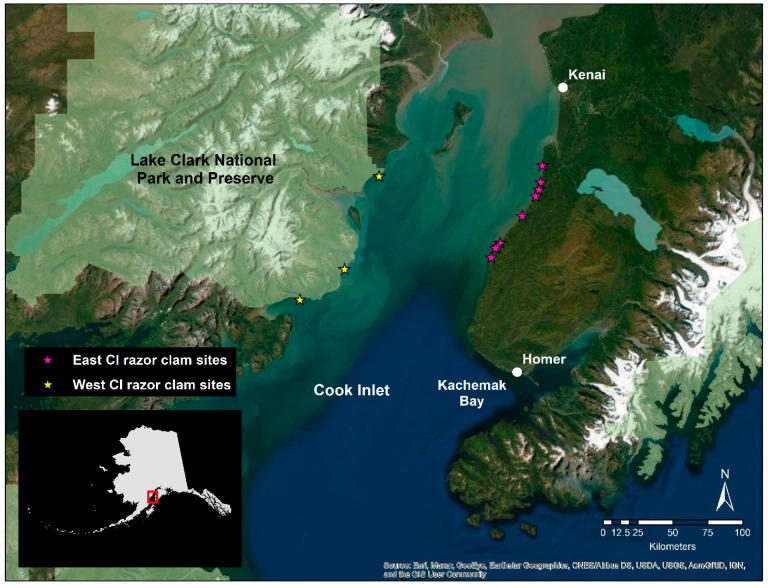
Razor clam sampling sites in ECI (pink stars) and WCI (yellow stars), Alaska. Sites were sampled during 2015 and 2016. ECI sites north to south: Cohoe, Clam Gulch North, Clam Gulch South, North Oil Pad Access, South Oil Pad Access, Ninilchik North, Ninilchik South, Ninilchik Bar, Deep Creek. WCI sites north to south: Polly Creek, Silver Salmon Creek, and Chinitna Bay.

**Figure 2 life-11-01288-f002:**
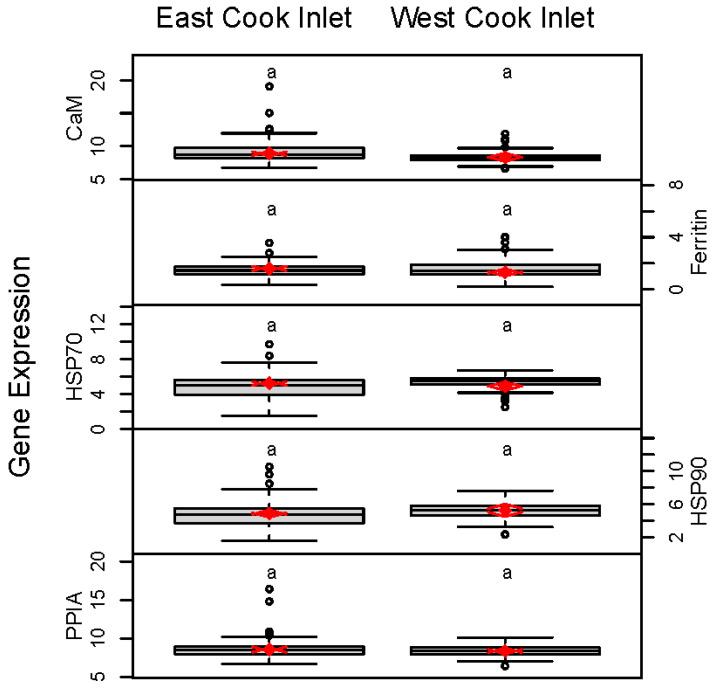
Region-level boxplots of data (C_T_ values) from five different gene expression assays performed on Pacific razor clams collected at three sites in WCI and nine sites in ECI. Random effects model results are denoted by red diamonds (mean) and red arrows (95% confidence intervals). Sites sharing a letter did not differ statistically based on post hoc testing. Analyzed by location (WCI vs. ECI), gene expression did not differ significantly. Note: smaller numbers indicate higher levels of expression.

**Figure 3 life-11-01288-f003:**
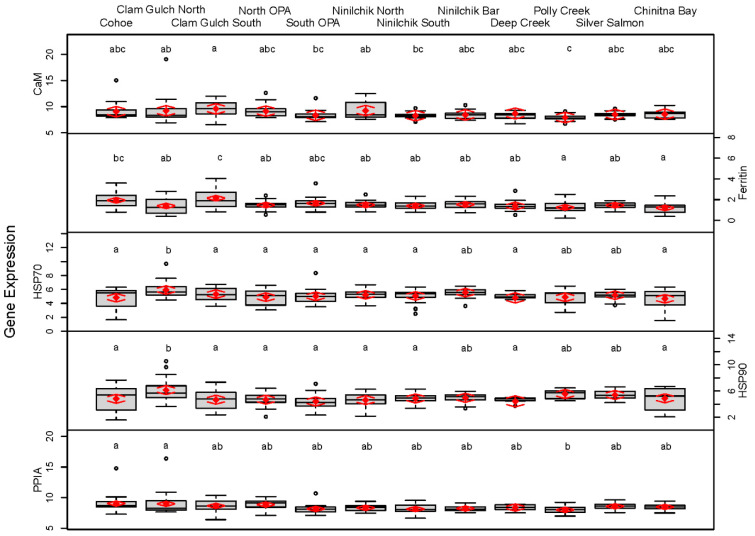
Site-level boxplots of data (C_T_ values) from gene expression assays on five genes, performed on Pacific razor clams collected at nine sites in ECI (Cohoe–Deep Creek) and three sites in WCI (Polly Creek–Chinitna Bay). Random effects of model results are denoted by red diamonds (mean) and red arrows (95% confidence intervals). Sites sharing a letter did not differ statistically based on post hoc testing. Note: smaller numbers indicate higher levels of expression. See Table 1 for gene names and functions.

**Figure 4 life-11-01288-f004:**
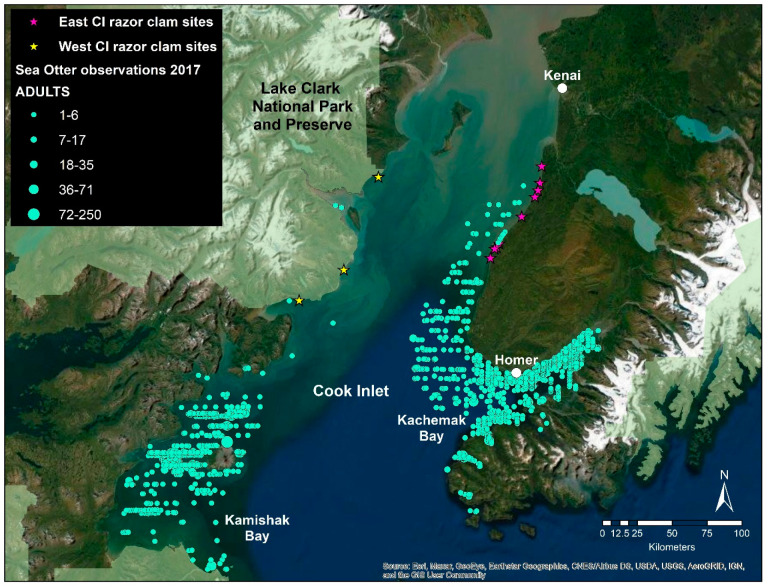
Sea otter abundance and distribution across Lower Cook Inlet, 2017 [23]. Sea otters are represented by light green dots. Dot size is representative of adult group size observed. East razor clam sites are represented by purple stars while west razor clam sites are represented by yellow stars.

**Table 1 life-11-01288-t001:** Genes selected to assess Pacific razor clam and ecosystem health.

Gene	Function
Calmodulin (CaM)	Shell formation—indication of ocean acidification; metabolism, immune response ([38]—Barnacle *Balanus amphitrite*; [39]—Grooved carpet shell clam, *Ruditapes decussatus*; [47]—Eastern oyster, *Crassostrea virginica*)
Ferritin (Ferr)	Increased in response to pathogens, metabolizing iron ([40]—Chinese razor clam *Sinonovacula constricta*; [41]—hard clam *Mercenaria*; [48]—zebra mussel *Dreissena polymorpha*)
Heat Shock Protein 70 (HSP70)	General stress, temperature, pathogen exposure, provides cellular protection ([43]—Mussels *Mytilus* spp.; [49]—Chinese razor clam *Sinonovacula contricta*)
Heat Shock Protein 90 (HSP90)	General stress, contaminants, temperature, salinity change, metabolism; provides cellular protection ([44]—black porgy *Acanthopagrus schlegeli*; [45]—Pacific oyster *Crassostrea gigas*; [46]—Zhikong scallop *Chlamys farreri;* [50]—short-necked clam *Paphia undulata*)
Peptidylprolyl isomerase A (PPIA)	Proinflammatory, increased in response to pathogen stimulus ([42]—Chinese mitten crab *Eriocheir sinensis*; [51]—moss animal *Fredericella sultana*)
18S	Reference ([52]—Chinese razor clam *Sinonovacula constricta*; [53]—Manilla clam *Ruditapes philippinarum* and grooved carpet clam *Ruditapes desussatus;* [54]—Chinese razor clam *Sinonovacula constricta*)
Elongation Factor Alpha-1 (EF1a)	Reference ([55]—soft-shell clams *Mya arenaria*; [56]—soft-shell clams *Mya arenaria*)

**Table 2 life-11-01288-t002:** Mean target gene expression levels (C_T_ values) followed by 95% confidence intervals in parentheses for sites in East (ECI) and West (WCI) Cook Inlet. Letters indicate statistical differences; sites sharing a letter did not differ statistically based on post hoc testing (see also Figure 3). Note: smaller mean values indicate higher levels of expression. A total of 20 samples were analyzed from each site except Deep Creek, which had only 10 samples collected in 2016. Gene names and functions are in Table 1.

Region	Site	CaM	Ferr	HSP70	HSP90	PPIA
ECI	Cohoe	9.03 (8.03, 10.02) abc	1.93 (1.67, 2.20) bc	4.81 (4.20, 5.42) a	4.81 (4.12, 5.49) a	9.04 (8.62, 9.46) a
	Clam Gulch North	9.21 (8.22, 10.20) ab	1.39 (1.13, 1.65) ab	5.91 (5.30, 6.52) b	6.13 (5.45, 6.82) b	8.98 (8.56, 9.40) a
	Clam Gulch South	9.58 (8.58, 10.57) a	2.14 (1.87, 2.40) c	5.33 (4.72, 5.94) ab	4.67 (3.99, 5.35) a	8.65 (8.24, 9.07) ab
	North Oil Pad Access	9.14 (8.14, 10.13) abc	1.48 (1.21, 1.74) ab	4.89 (4.28, 5.50) a	4.68 (3.99, 5.36) a	8.86 (8.44, 9.28) ab
	South Oil Pad Access	8.34 (7.35, 9.34) bc	1.64 (1.38, 1.91) abc	5.01 (4.40, 5.62) ab	4.40 (3.72, 5.09) a	8.16 (7.74, 8.58) ab
	Ninilchik North	9.25 (8.26, 10.25) ab	1.49 (1.23, 1.75) ab	5.23 (4.62, 5.83) ab	4.61 (3.92, 5.29) a	8.33 (7.91, 8.75) ab
	Ninilchik South	8.28 (7.28, 9.27) bc	1.40 (1.14, 1.66) ab	5.07 (4.46, 5.68) ab	4.85 (4.17, 5.54) a	8.13 (7.71, 8.55) ab
	Ninilchik Bar	8.43 (7.43, 9.42) abc	1.53 (1.27, 1.79) ab	5.54 (4.93, 6.15) ab	4.95 (4.27, 5.64) ab	8.22 (7.81, 8.64) ab
	Deep Creek	8.68 (7.63, 9.74) abc	1.38 (1.02, 1.75) ab	4.78 (4.06, 5.51) ab	4.42 (3.57, 5.27) a	8.38 (7.79, 8.97) ab
	All ECI	8.87 (8.64, 9.10)	1.61, (1.51, 1.71)	5.21 (5.05, 5.37)	4.87 (4.67, 5.07)	8.54 (8.37, 8.70)
WCI	Polly Creek	7.92 (6.92, 8.91) c	1.25 (0.98, 1.51) a	4.87 (4.26, 5.48) a	5.54 (4.85, 6.22) ab	8.00 (7.58, 8.42) b
	Silver Salmon	8.49 (7.50, 9.49) abc	1.46 (1.20, 1.72) ab	5.17 (4.56, 5.78) ab	5.39 (4.70, 6.07) ab	8.58 (8.16, 9.00) ab
	Chinitna Bay	8.55 (7.56, 9.55) abc	1.20 (0.93, 1.46) a	4.66 (4.05, 5.27) a	4.88 (4.19, 5.56) a	8.48 (8.06, 8.90) ab
	All WCI	8.32 (8.15, 8.49)	1.30 (1.17, 1.43)	4.90 (4.63, 5.17)	5.27 (4.98, 5.56)	8.35 (8.20, 8.50)

## Data Availability

Data available upon request.

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
