# Peer review of "Gene Expression Profiles in Two Razor Clam Populations: Discerning Drivers of Population Status"

_life, 2021, doi:10.3390/life11121288_

Round 1

Reviewer 1 Report

The manuscript titled as “Gene transcription profiles in two razor clam populations: discerning drivers of population status” describes the research work to finding the causes of population differences. The conclusion is interesting that no differences of gene transcription profiles and physiological processes between the two razor clam populations, so that the real cause should be considered from the other factors such as predation or changes in habitat. This is a real problem driving study trying by applying the gene transcription profiles, however unexpectely showing negative results. It presents some direct view to think about the important drivers for the population changes should be more concerned about the habitat interactions such as prey and predator relations.   

Author Response

We appreciate Reviewer 1's comments and support. We made some minor revisions to the manuscript in response to Reviewer 3, primarily to provide more information about the razor clam fishery prior to closure. However, the intent of the manuscript and its conclusions have not changed. 

Thank you,

Heather Coletti (lead author)

Reviewer 2 Report

This research is interesting and meaningful for guiding the population recovery of razor clam Siliqua patula in East Cook Inlet. Although authors used representative genes to distinguish the physiological differences between those two populations (WCI and ECI), other potential genes might also exist playing critical roles. Thus, transcriptome data might be helpful and comprehensive to clarify the genetic factors influencing the population difference. Nevertheless, more attentions should be given the extrinsic factors to help the razor clam flourishing in ECI. In general, this article is well presented, and could be accepted in your journal of “Life”.

Author Response

We appreciate Reviewer 2's comments and support. We agree that to improve our ability to use transcriptomics to attribute specific drivers, controlled laboratory exposure studies with whole transcriptome responses are needed. These controlled studies have been proposed and funded for summer 2022 in partnership with the Alaska SeaLife Center. Outcomes from that work will be informative in addition to further studies on predation pressure to determine drivers of population status. 

We made some minor revisions to the manuscript in response to Reviewer 3, primarily to provide more information about the razor clam fishery prior to closure. However, the intent of the manuscript and its conclusions have not changed. 

Thank you,
Heather Coletti (lead author)

Reviewer 3 Report

The authors are using a differential gene expression analysis to determine whether or not site specific population declines may be attributed to unforeseen stressors following a commercial fishing ban. The long-term impacts of fishing 1 million individuals annually from 1977 - 2006, with notable declines in populations starting around 2009. The eastern of the Cook Inlet is easily accessible and near highly populated areas, whereas the western shore is only accessible by boat and not near a more highly populated area.

The authors use a targeted gene approach rather than overall comparative transcriptomics. This approach was used because of previously identified stress markers (Bowen et al. 2020) with one notable difference: The previous study sampled hemolymph, while the current study sampled the gills. The notable difference in source RNA may be attributed to the authors not finding any significant differences in the focal populations, but the authors did not acknowledge this in their current manuscript and should cite sources if an alternative tissue type will provide the same stress indicators. Additionally, the authors potentially attribute the difference in populations to either undetectable physiological factors or unaccounted predators. Although these are both possible, it seems to me that it's more likely that the razor clam population was harvested to extreme levels, so much so that after several decades of 1 million animals/year they are still recovering. If this is not the case the authors need to more details with regards to sustainable harvesting of razor clams and expand on how the west coast population of the Cook Inlet doesn't seem to be impacted. To me it seems simply their isolation saves them from overharvesting. Has the west coast been under the same harvesting pressures? If not how does it compare over the years to the east coast?

I have attached additional comments with regards to editing/improvement. Although I did not have enough time to provide as much feedback in this area as I would like to with this regard during this round of reviewing, I would be happy provide additional comments during the second round after the authors are able to consider: tissue specific differences as it relates to their interpretation and how long-term overfishing may have contributed to this.  

Author Response

We appreciate Reviewer 3 noting the possible point of confusion (gill tissue and hemolymph) and added slight modifications to the text to clarify that gill tissue was used in transcriptome analysis in both studies, Bowen et al. 2020 and this paper. Gill tissue and not hemolymph was used for all gene transcription analyses in Bowen et al. (2020). Please see Invertebrate gene transcription section in Bowen et al. (2020). Hemolymph was only used for physiological assays including hemocyte count and hydrogen peroxide production, not gene transcription.

We also appreciate the reviewer’s thoughts on human harvest pressure and have added text to the paper that we hope clarifies this issue. In summary, Alaska Department of Fish and Game (ADF&G) has long monitored the population of razor clams and of razor clam harvest rates on the east side of Cook Inlet. Razor clam harvest rates were not evenly distributed throughout the area and primarily occurred on the Clam Gulch and Ninilchik area beaches. Between 1977 – 2006, the fishery remained stable with consistent recruitment of new age classes to the beaches and harvest was comprised of a broad range of age classes on all beaches (Szarzi et al. 2010 ). However, between 2009 and 2012, the annual number of clams harvested per digger declined by 41% concurrent with a decline in harvest effort (fewer people clamming) below the long-term mean (1977-2008). ADF&G concluded that there was a dramatic decline in the ECI razor clam population. Evidence to support that this was not due to overharvest comes from ADF&G’s continual monitoring of the clam population and its lack of recovery in the absence of harvest. ADF&G has found that recruitment of juveniles into the population remained stable during closure. However, average annual age and length compositions were truncated to younger/smaller clams compared to historical averages regardless of the decreased harvest rates (closures). The well below historical average densities and abundances of juvenile clams in almost all study areas suggests that clams failed to recruit to adult age classes in these study areas (Kerkvliet et al. 2021).

In addition, the exploitation rate of razor clams by humans throughout most of ECI was assumed low based on digger distribution and digger monitoring at more heavily harvested beaches at Clam Gulch and Ninilchik (Szarzi and Hansen 2009 ). Even with the partial closure in 2013 and full closure in 2015, the ECI razor clams still have not recovered to historical abundances. The causes of the decline were unknown but were attributed to poor recruitment into adult age classes and above average adult natural mortality (Kerkvliet et al. 2016). The dramatic drop in adult clam abundances at the Ninilchik South study area from 2011 through 2015 highlights a high natural mortality rate.

In contrast, the razor clam fisheries in WCI have continued to support a commercial and personal use fishery over the same time frame. Essentially, clams aren’t being allowed to grow into larger (harvestable) size classes across ECI, which points to predation as opposed to over-harvest, since the harvest has been closed for several years now. Recruitment of juveniles into the population is still good. We have added text and additional citations to the paper that we hope clarifies this point.

We have also addressed some of the editorial comments and appreciate the reviewer’s attention to detail. The attached manuscript has the track changes embedded for consideration.

Thank you,

Heather Coletti (lead author)

Round 2

Reviewer 3 Report

The authors have adequately clarified points of concern identified in the my previous review and I appreciate these clarifications and expanded material in the introduction. I appreciate brining to my attention the miss-alignment of hemolymph and gills in my previous comments. I see now that gills are referenced in Bowen et al. 2020. It seems the next reasonable step would be comparative transcriptomics using high throughput sequencing or lab manipulations with targeted genes of interest. Although the reference to having non-significant important results (lines 377 - 379) is a very important statement and of interest to the broader readership in this journal.

There are still some minor writing style changes/overall fluidity of the writing that I would suggest the authors consider:

  • Lines 20 - 23
  • Lines 25 - 26
  • Lines 269 - 271

Author Response

We appreciate Reviewer 3's continued attention to detail and suggestions for improvement. We made slight modifications to improve readability where as suggested. We appreciate the recognition that a lack of differences can be a significant finding important to ecological studies. 

Thank you, 

Heather Coletti (lead author)